# Transcriptome analysis reveals Nitrogen deficiency induced alterations in leaf and root of three cultivars of potato (*Solanum tuberosum* L.)

**Jingying Zhang**◉, **Yaping Wang**◉, **Yanfei Zhao, Yun Zhang, Jiayue Zhang, Haoran Ma, Yuzhu Han**◉*

College of Horticulture, Jilin Agricultural University, Changchun City, P.R. China

◉ These authors contributed equally to this work.
* hanyuzhu@jlau.edu.cn

**Data Availability Statement:** All relevant data are within the paper and its Supporting Information

## Abstract

Nitrogen (N) is a key element for the production of potato. The N uptake efficiency, N use efficiency and increased N utilization efficiency can be decreased by N deficiency treatment. We performed this study to investigate the association between transcriptomic profiles and the efficiencies of N in potato. Potato cultivars "Yanshu 4" (short for Y), "Xiabodi" (cv. Shepody, short for X) and "Chunshu 4" (short for C) were treated with sufficient N fertilizer and deficient N fertilizer. Then, the growth parameters and tuber yield were recorded; the contents of soluble sugar and protein were measured; and the activities of enzymes were detected. Leaf and root transcriptomes were analyzed and differentially expressed genes (DEGs) in response to N deficiency were identified. The results showed that N deficiency decreased the nitrate reductase (NR), glutamine synthetase (GS) and root activity. Most of the DEGs between N-treated and N-deficiency participate the processes of transport, nitrate transport, nitrogen compound transport and N metabolism in C and Y, not in X, indicating the cultivar-dependent response to N deficiency. DEGs like glutamate dehydrogenase (*StGDH)*, glutamine synthetase (*StGS)* and carbonic anhydrase (*StCA)* play key roles in these processes mentioned above. DEGs related to N metabolism showed a close relationship with the N utilization efficiency (UTE), but not with N use efficiency (NUE). The Major Facilitator Superfamily (MFS) members, like nitrate transporter 2.4 (*StNRT2.4)*, 2.5 (*StNRT2.5)* and 2.7 (*StNRT2.7)*, were mainly enriched in the processes associated with response to stresses and defense, indicating that N deficiency induced stresses in all cultivars.

## 1. Introduction

Potato (*Solanum tuberosum* L.) is the most important non-grain food crop in the world, the 4th most important staple food crop and the 12th most important agricultural product [1]. Potato is rich in carbohydrates, vitamins, minerals and proteins that are essential for humans,

files. Sequence data were available in SRA database with the number of SRS4186597.

**Funding:** The authors are grateful to Jilin Agricultural University Comprehensive Experimental Platform. This research was funded by Jilin development and Reform Commission Project (20160601) and Project of science and Technology Department of Jilin Province (20180201001NY).The funders had no role in study design, data collection and analysis, decision to publish.

**Competing interests:** The authors have declared that no competing interests exist.

**Abbreviations:** BPs, biological processes; C, "Chunshu 4"; dsDNA, double strand DNA; DEG, differentially expressed gene; FPKM, fragments per kilobase of exon model per million mapped reads; GDH, glutamate dehydrogenase; GO, Gene Ontology; GS, glutamine synthetase; KEGG, Kyoto Encyclopedia of Genes and Genomes; N, Nitrogen; NiR, nitrite reductase; NR, nitrate reductase; NRTs, nitrate transporters; NUE, N use efficiency; TTC, triphenyl tetrazolium chloride; UPE, N uptake efficiency; UTE, N utilization efficiency; X, "Xiabodi"; Y, "Yanshu 4".

its tuber is used as a staple food globally [1–3]. Nitrogen (N) is a necessary nutrient for crop growth, especially for the production of N-sensitive crop potato [1–4]. For potato pro duction, N content in soil is a limiting factor. Therefore, the utilization efficiency of N is extremely important for the growth of potatoes. N efficiency estimation has been widely used to measure the capacity of plants to acquire and utilize nutrients for biomass production [5]. The N use efficiency (NUE, calculated as the tuber yield per unit of nutrient supply from soil and fertilizer in potato) is an important index for the production and market value of potato. As an important agronomy practice, applying nitrogen fertilizer in potato cropping can greatly improve potato production [6]. Therefore, understanding the N responsiveness in potato is important for high-NUE potato varieties breeding. The ideal potato genotype has both high genetic NUE and high N reactivity [7,8].

Improving NUE is one of the most effective means to increase crop productivity while reducing environmental degradation and farmers' costs [9]. NUE could be divided into N uptake efficiency (UPE), and N utilization efficiency (UTE, calculated as the tuber yield divided by the maximum plant N pool) [10]. The UTE represents the efficiency of assimilation and remobilization of plant N to ultimately produce grain [11]. Improving UTE can reduce N consumption, maintain (even increase) production and may reduce excessive input of nitrogen fertilizer. UTE ranges from 50% to 80% in potato, and may be affected by many factors, such as N fertilizer application, agronomic measures and plant genetic factors [9,12]. In order to use nitrogen more efficiently and economically, different approaches had been tried to reduce the use of N fertilizers while ensuring yield [13].

There are many agricultural practices in improving NUE and crop yield, including controlling plant density, sowing time, and breeding high-NUE plants [12,13]. For instance, Yin et al demonstrated that delaying sowing and reducing N fertilizer application could achieve comparable yields of wheat [9]. They reported that delaying sowing time could decrease the UPE and spike density, and increase the UTE and grain number, but not effects found on grain yields [9]. In addition to agronomic strategies, plant genetics information is also an important factor. There are various genes associated with N absorption and utilization, like nitrate transporters (*NRTs*), nitrate reductase (*NR*), glutamine synthetase (*GS*), glutamate dehydrogenase (*GDH*) and nitrite reductase (*NiR*) [12]. It has been reported that the GS overexpressing plant can increase the UTE and UPE in wheat and barley, which results in increasing crop yields [14,15]. Understanding the relationship of gene expression pattern and NUE is important for potato breeding.

In the production practice, we found that the utilization efficiency of N of "Yanshu 4", "Xiabodi" (cv. Shepody) and "Chunshu 4" potatoes are quite different in the production practice. According to production experience, under the condition of sufficient nitrogen fertilizer, "Yanshu 4" was a high-absorption but low-utilization potato, "Xiabodi" was a medium-absorption and medium-utilization potato, and "Chunshu 4" was a low-absorption but high-utilization potato. In this study, we performed this study to investigate the difference in NUE, UPE, UTE and gene expression profiles among these three potato cultivars. All these three potato cultivars were applied with N-complete and N-deficient fertilizer. The growth, production and physiology parameters were detected and analyzed in combination with transcriptome analysis. This study might add more information on N deficiency-induced molecular profiles.

## 2. Materials and methods

### 2.1. Plant and experiment design

Three potato cultivars "Yanshu 4" (Y for short), "Xiabodi" (cv. Shepody, X for short) and "Chunshu 4" (C for short) were planted into pots in an artificial climate chamber with

conditions of 24˚C, 60% humidity and 14:10 light/dark cycle. All potato tubers were obtained from the planting Resource Bank of Jilin Agricultural University. The potato plants of each cultivar were randomly divided into two groups: N-complete group (short for N) and N-deficient nutrient group (30 plants per group). Plants in N group were routinely treated with 3.3kg/100 m$^2$ N (Urea), 1.8kg/100 m$^2$ P (P$_2$O$_5$), and 2.7 kg/100 m$^2$ K (K$_2$O); Plants in N-deficient nutrient group were treated with 1.8kg/m$^2$ P(P$_2$O$_5$) and 2.7kg/100m$^2$ K (K$_2$O), and no N fertilizer applied. The amount of fertilizer is converted according to the field production practice. Accordingly, samples were assigned into 6 groups according to the treatment and cultivar difference, including "Yanshu 4" treated with (short for YN) and without N (short for Y); "Xiabodi" treated with (short for XN) and without N (short for X); "Chunshu 4" treated with (short for CN) and without N (short for C). Y, X and C indicate the cultivar of "Yanshu 4", "Xiabodi" and "Chunshu 4", respectively. N notes treatment with 22 kg/667 m$^2$ N. Each group was repeated for 4 times.

## 2.2. Measurement of physiological and biochemical parameters

Nine plants in each group were randomly selected, three of which were pooled as a repeat and a total of three replicates were set for each group. The fresh and dry weight (g/plant) of all the leaves in a plant were recorded at the mature stage. Also, the fresh and dry weight (g/plant) of stem and root of each plant were measured. Also, the total fresh tuber of each plant was weighted by a platform scale. The anthrone method [16] and Coomassie Brilliant Blue G-250 dye-binding method [17] were used to detect the contents of soluble sugars (carbohydrate) and soluble proteins of the flag leaf, respectively. The data of five developmental stages (seedling, bud, tuber bulking, starch accumulation and mature stages) were recorded continuously to evaluate the soluble sugars and soluble proteins variation. The activity of nitrate reductase (NR, EC 1.7.1.3) and glutamine synthetase (GS, EC-6.3.1.2) were detected using a glutamine synthetase assay kit (Nanjing Jiancheng, Bioengineering Institute, China) according to the manufacturer's instructions. Briefly, the enzyme solution of plant tissues was extracted with sodium phosphate buffer. Then the color reaction was carried out and measured by a Microplate ELISA Reader (BioTek, USA). Then the enzyme activity was calculated based on the manufacturer's instruction. Root activity directly affects the nutritional status and yield of the aboveground. Root activity was measured using triphenyl tetrazolium chloride (TTC) method reported previously [18]. Briefly, 0.5g root samples submitted to different ClO$_2$ concentrations, immersed in a 10ml beaker with 0.4% TTC and 66mM sodium phosphate buffer. Samples were then put into graduated test tubes filled with10ml of methanol. Afterwards, the test tubes were left at 37˚C for 4-7h in an incubator until the apical section turned completely white. Using a spectrophotometer for 485nm colorimetry. Root activity = amount of TTC reduction (μg)/fresh root weight (g) × time (h). Total N content in plant leaf, stem, root and soil was determined using an automatic kjeldahl apparatus (UDK159, VELP scientifica, Italy). N-efficiency parameters including NUE, UPE and UTE was calculated according to Zareabyaneh's report [19].

## 2.3. RNA-seq analysis

Nine plants at bud stage in each group were randomly selected, three of which were pooled as a repeat and a total of three replicates were set for each group. The second leaf (the leaf under flag leaf) and root tissues were sampled for RNA-seq analysis using Illumina HiSeq 4000 platform. Three leaf (a) and root (b) samples were obtained from "X", "Y" and "C". Total RNA in each sample was purified using the cetyltrimethylammonium bromide (CTAB) method according to Thunyamada's report [20]. Genomic DNA contamination were removed by

DNase I (Takara, Tokyo, Japan). RNA concentration was determined using a NanoDrop ND-2100 spectrophotometer (NanoDrop Technologies, Wilmington, DE, USA). Then, RNA sequencing library was constructed using a mRNA-seq Library Prep Kit for Illumina (Vazyme, Nanjing, Jiangsu, China). All the RNA sequencing libraries were then detected using a Qubit 2.0 fluorometer (Invitrogen, Carlsbad, CA, USA). In addition, the integrity of RNA was also determined by an Agilent 2100 Bioanalyzer (Agilent Technologies, Santa Clara, CA, USA). All these libraries were then subjected to the Illumina HiSeq 4000 platform followed by $2 \times 150$ bp paired-end sequencing.

## 2.4. Data processing and gene expression profile

Illumina CASAVA software (version 1.8.2, Illumina, Hayward, CA, USA) was used for converting the original image data to sequences by base calling. FastQC (version 0.11.5, http://www.bioinformatics.babraham.ac.uk/projects/fastqc/) was used for the quality control of raw data in the format of fastq. The adaptor and low-quality reads were then removed. HISAT software (version 2.0.4) [21] was used for mapping the clean data to the reference genome. Transcript assembly was conducted using cufflinks (version 2.1.1) [22], and the novel genes were identified using Cuffcompare in cufflinks (version 2.1.1) [22]. The fragments per kilobase of exon model per million mapped reads (FPKM) values of each read was calculated and the differentially expressed genes (DEGs) between two groups were identified using DESeq (version 1.12.0) [23] with the criteria of padj < 0.05.

## 2.5. Enrichment analysis

The functional categories associated with DEGs were identified based on the enrichment analysis. Gene Ontology (GO) biological processes (BPs) associated with DEGs were identified using GOseq [24]. KEGG (Kyoto Encyclopedia of Genes and Genomes) pathway enrichment analysis was performed using the KOBAS (version 2.0) [25]. Terms with padj < 0.05 were considered as significant enrichment.

## 2.6. qRT-PCR analysis

The relative expression of several DEGs related to N metabolism were validated by qRT-PCR analysis. At bud stage, leaf (second leaf) and root RNA was isolated using the CTAB method [20], and cDNA were synthesized as aforementioned. The relative expression were detected using the SHRR GREEN Mastermix (TaKaRa, Japan) and StepOnePlusTM Real-Time PCR System (Applied Biosystems, USA). The specific primers of the related genes were listed in S1 Table. The fold change of mRNA was calculated using the $2^{-\Delta\Delta Ct}$ method by normalizing to the internal control gene EF1α.

## 2.7. Statistical analysis

Data were expressed as the mean ± SD (n = 3) and were subjected to the statistical analysis using GraphPad Prism 6 software. Statistical differences between two groups were analyzed using unpaired *t*-test, and that among six groups that were analyzed using the two-way ANOVA test followed by Tukey post-hoc. Correlation were analysis by Pearson correlation analysis. A probability value of less than 0.05 was considered a significant difference.

## 3. Results

### 3.1. N deficiency decreases potato growth

As expected, N-deficient fertilization significantly decreased the fresh leaf weight (by about 40%), dry leaf weight (by about 20%), stem weight (by about 34%) and fresh tuber weight (by about 16%) in all these three potato varieties ($p < 0.01$, Fig 1). We also found that the N ratio in leaf was significantly increased by N deficiency treatment (about 18%, p<0.01, Fig 2), but decreased in root and stem. The N contents (g/plant) in leaf, stem and root were lower in N-deficient group than that in N group. In addition, the UPE, UTE and NUE indexes were also significantly decreased by about 17%, 8% and 7% respectively by N deficiency in all these three potato varieties (Fig 2).

### 3.2. Influences of N deficiency on quality properties and enzyme activities

The results showed that the N deficiency treatment significantly reduced the contents of soluble sugars and proteins in the leaves, but the effect was different in different developmental stages (Fig 3A and 3B). After N-deficiency treatment, cultivar C had the lowest soluble sugar (2.31±0.08, 0.76±0.03 and 0.76±0.03) and protein contents (1.18±0.03, 1.22±0.03 and 0.98 ±0.03) at the bud, tuber bulking and starch accumulation stages. Also, the root and GS activity of plants in Y (49.91±1.18 and 0.12±0.01), X (71.68±5.21 and 0.29±0.01) and C (63.78±5.21 and 0.23±0.01) at the seedling, tuber bulking and starch accumulation stages were significantly lower than that in Yn (72.37±5.21 and 0.18±0.01), Xn (96.00±2.26 and 0.33±0.01) and Cn (81.00±1.91 and 0.25±0.01), respectively (Fig 3C and 3E). Interestingly, N-deficient fertilization treatment significantly increased the NR activity (by about 25%) at the seedling stage, but decreased it (by about 16%) at the tuber bulking, starch accumulation and mature stages (Fig 3D). At the mature stage, the soluble sugars and proteins contents in leaf were low, which were not dependent on the application of N fertilizer, but on cultivars. N deficiency also decreased the NR (by ~16.55%) and GS activities (by ~15.06%) in potato leaves at the mature stage (Fig 3D and 3E).

### 3.3. Summary of illumina sequencing

Illumina sequencing generated 1535.22 M clean reads and 241.49 G bases, with an average GC content and Q30 value of 41.45% and 93.84%, respectively (S2 Table). We identified 18, 90 and 1446 DEGs in the leaf samples of cultivar C (Ca vs. CNa), X (Xa vs. XNa) and Y (Ya vs. YNa),

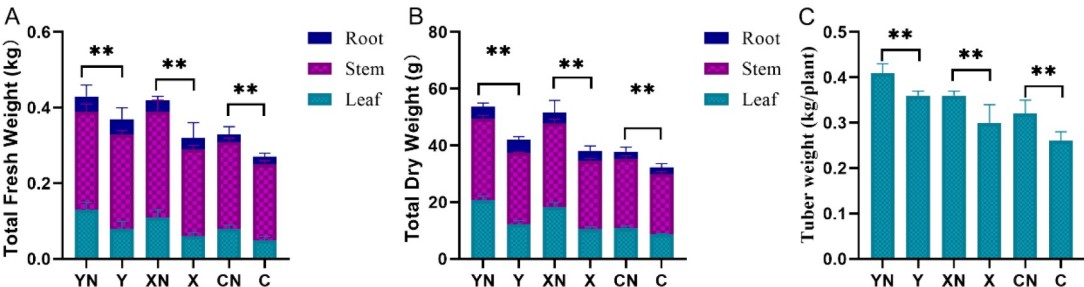

**Fig 1. Statistical analysis of the growth performance parameters in response to by nitrogen (N) deficiency.** A, B and C shows the total fresh weight, total dry weight and tuber weight, respectively. N = 3, two-way ANOVA test followed by Tukey post-hoc, ** P <0.01 vs. corresponding variety under N treatment. Y, X and C indicate the cultivar of "Yanshu 4", "Xiabodi" and "Chunshu 4", respectively. N means treatment with N containing fertilizer (control). For example, YN and Y means "Yanshu 4" treated with N-containing and N-deficiency fertilizer, respectively.

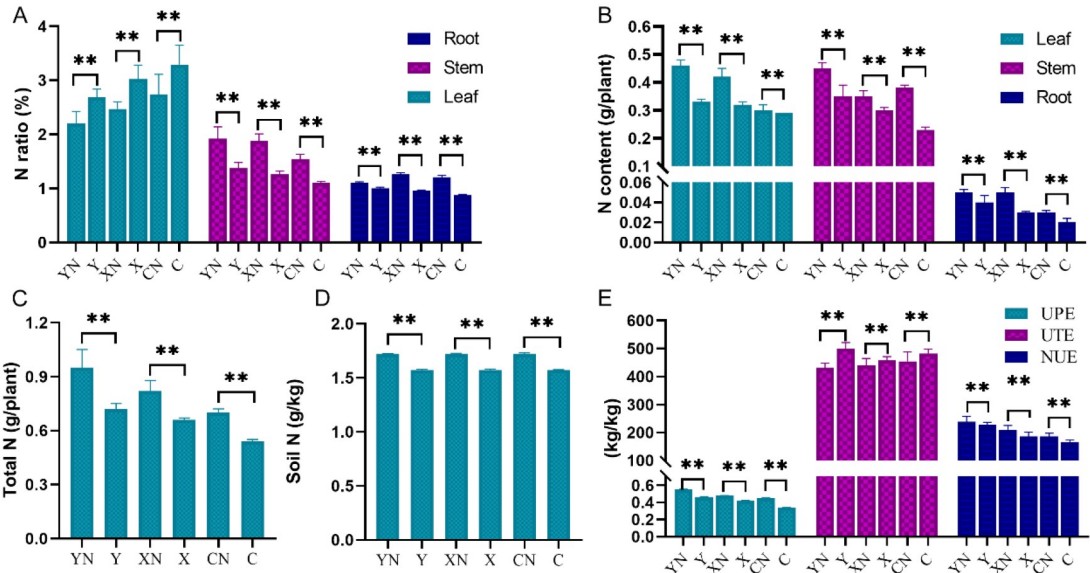

**Fig 2. Statistical analysis of the biochemistry indices of potato plant and soil in response to N deficiency.** A and B shows the N ratio (%) and N content (g/plant), respectively. C and D shows the total N (g/plant) and soil N (g/kg), respectively. E shows the value of NUE, UTE and UPE. NUE, N use efficiency; UPE, N uptake efficiency; UTE, N utilization efficiency; N = 3, two-way ANOVA test followed by Tukey post-hoc, ∗∗ P <0.01 vs. corresponding variety under N treatment. Y, X and C indicate the cultivar of "Yanshu 4", "Xiabodi" and "Chunshu 4", respectively. N means treatment with N containing fertilizer (control). For example, YN and Y means "Yanshu 4" treated with N-containing and N-deficiency fertilizer, respectively.

respectively. For root samples, there were 0, 0 and 1009 DEGs found in cultivar C (Cb vs. CNb), X (Xb vs. XNb) and Y (Yb vs. YNb), respectively (Fig 4A and 4B). In leaves, the number of DEGs of pairwise comparison between varieties were showed in Fig 4C and 4E. The number of DEGs in root of pairwise comparison between varieties were showed in Fig 4D and 4F. Overall, N-deficiency treatment reduced the number of DEGs in the comparisons among different cultivars (Fig 4C and 4D).

## 3.4. N deficiency-induced DEGs associated with the N metabolism

The annotation results showed that most of the DEGs between CN and C were related to the processes of transport, nitrate transport and nitrogen compound transport (Fig 5A). *StNRT 2.5* (PGSC0003DMG400016996), which was down-regulated in the leaf by N deficiency, act as a key regulator in variety C (Fig 5B). GO enrichment analysis showed that DEGs between YN and Y were significantly enriched in biological processes associated with N metabolism, including organonitrogen compound catabolic process, glutamine family amino acid metabolic process, arginine catabolic process to glutamate, and glutamate metabolic process (Fig 5C and S3 Table). DEGs like PGSC0003DMG400008356 (*StGDH*), PGSC0003DMG400023620 (*StGS*) and PGSC0003DMG400030984 (*StCA*) play key roles in these processes mentioned above. N deficiency treatment significantly decreased the expression of *StGDH* and *StGS* in the root of Y (Fig 5D and 5E). While *StGS* was increased by N deficiency treatment in leaves (Fig 5F). No N metabolism related GO terms were significantly enriched in X.

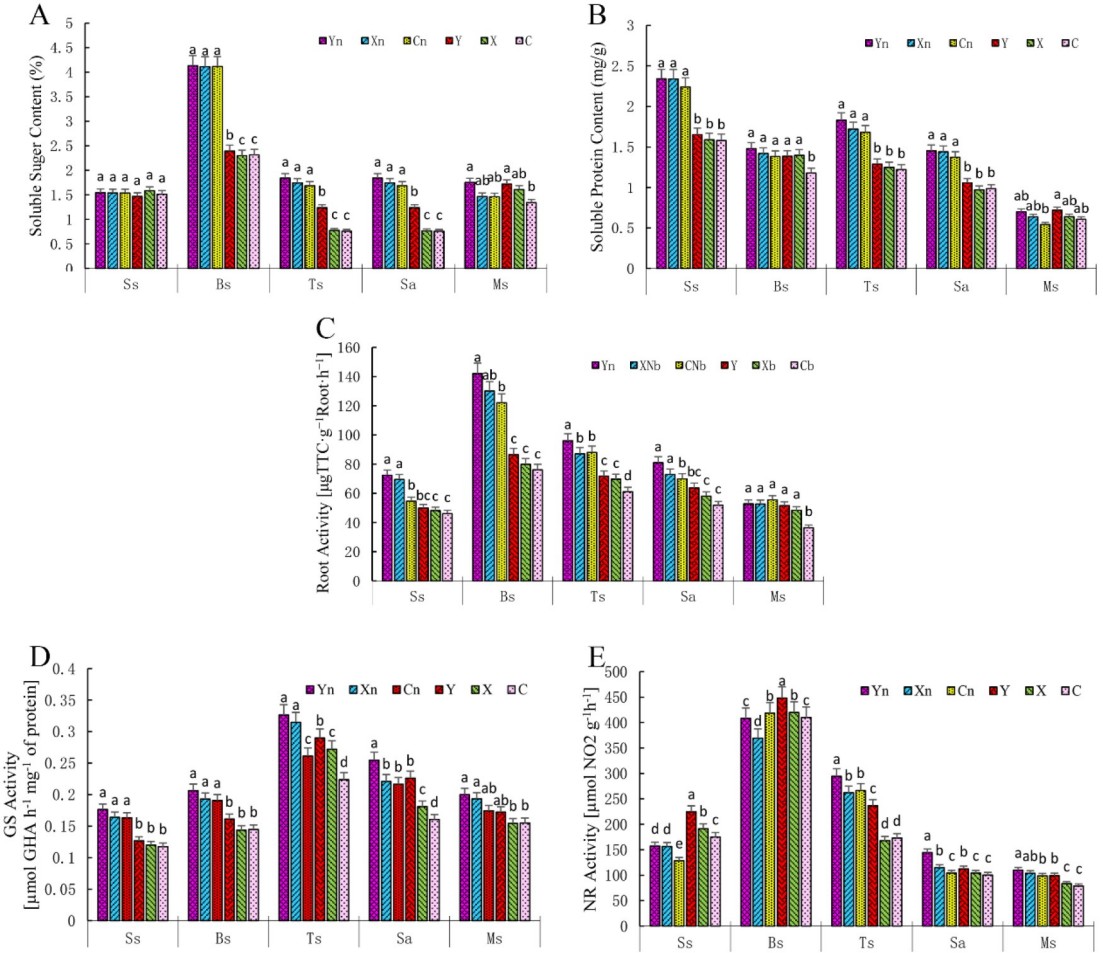

**Fig 3. Influence of N-deficiency fertilization on potato quality and enzyme activity.** (A), Soluble sugar content; (B), soluble protein content; (C), root activity; (D), NR activity; (E), GS activity. NR, nitrate reductase; GS, glutamine synthetase. Ss, Bs, Ts, Sa and Ms indicates the seedling, bud, tuber bulking, starch accumulation and mature stage, respectively. For statistical analysis, n = 3, two-way ANOVA test followed by Tukey post-hoc. Different letters indicate that mean values are significantly different (p<0.05). Y, X and C indicate the cultivar of "Yanshu 4", "Xiabodi" and "Chunshu 4", respectively. N means treatment with N containing fertilizer (control). For example, YN and Y means "Yanshu 4" treated with N-containing and N-deficiency fertilizer, respectively.

### 3.5. DEGs related to N metabolism among cultivars and between leaf and root

By comparing the transcriptome in different cultivars, many DEGs were identified among different cultivars after N containing fertilizer treatment (Fig 4C and 4E). Under N containing fertilization conditions, the up-regulated DEGs between XNa vs CNa were significantly enriched in "Photosynthesis", "Porphyrin and chlorophyll metabolism", "Carbon metabolism" and "Biosynthesis of secondary metabolites" (Table 1). While the down-regulated DEGs between XNa vs CNa were significantly enriched in "Mismatch repair", "DNA replication", "Plant hormone signal transduction", "Nicotinate and nicotinamide metabolism" and "Valine, leucine and isoleucine degradation" (Table 1).

Moreover, several N metabolism related DEGs were identified. For instance, the expression of ferredoxin-nitrite reductase (*StNiR*) gene in X were significantly lower than that in C in

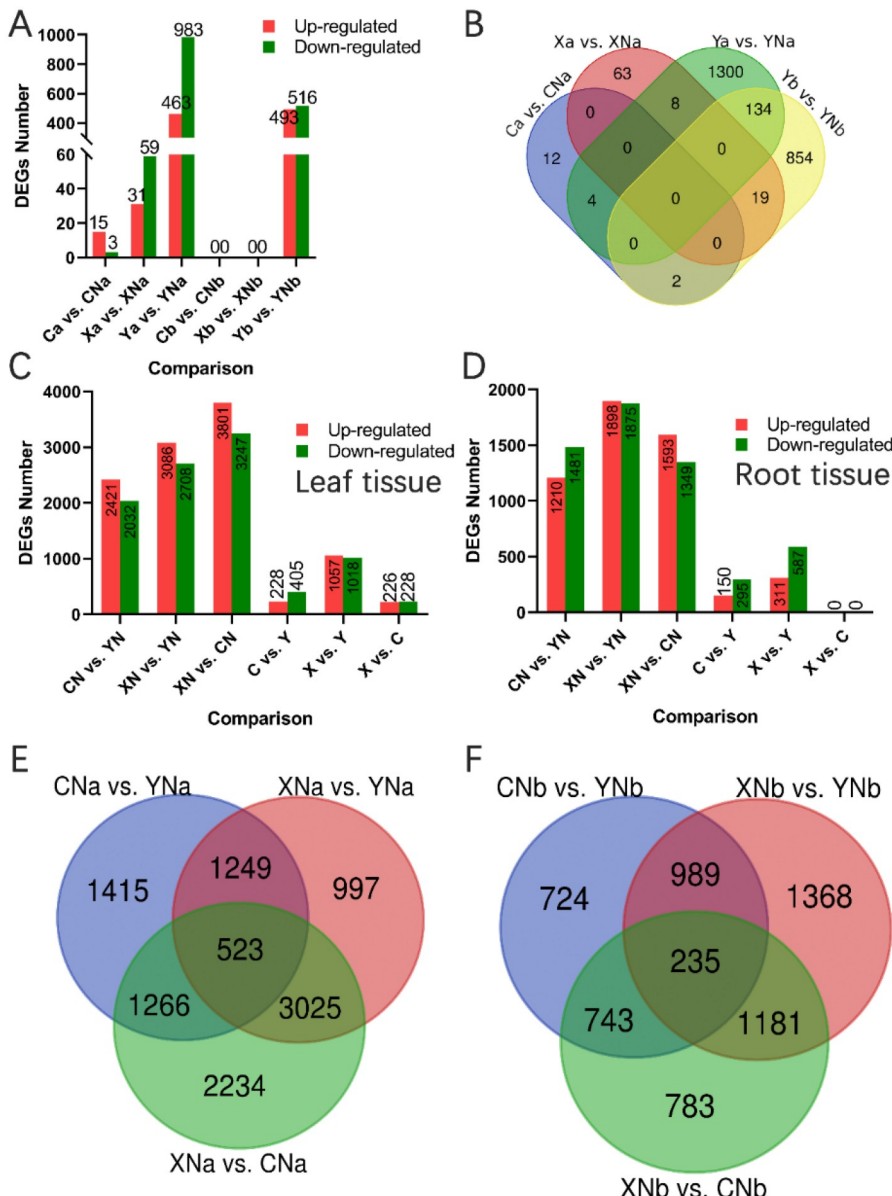

**Fig 4. Differentially expressed genes (DEGs) between different treatments.** A, the number of DEGs in the leaf (a for short) and root (b for short) induced by N deficiency. B, the Venn diagram of the DEGs in the leaf of potato in response to N deficiency. C and D, the statistics and Venn diagram of the DEGs in leaves of potato, respectively. E and F, the statistics and Venn diagram of the DEGs in root of potato, respectively. The red bar indicates up-regulated genes; green bar indicates down-regulated genes. A and b are short for leaf and root, respectively. Y, X and C indicate the cultivar of "Yanshu 4", "Xiabodi" and "Chunshu 4", respectively. N means treatment with N containing fertilizer (control). For example, YN and Y means "Yanshu 4" treated with N-containing and N-deficiency fertilizer, respectively.

both the leaf and root tissues (Fig 6). In addition, many major facilitator superfamily (MFS) members were clustered, which showed different expression patterns between root and leaf in Y (Fig 6). Among these MFS members, there were several high affinity *StNRTs*, including *StNRT2.4*, *StNRT2.5* and *StNRT2.7* (Fig 6). These multiple expression profiles of MFS members suggested the nonredundant functions in potato leaf and root. However, the number of

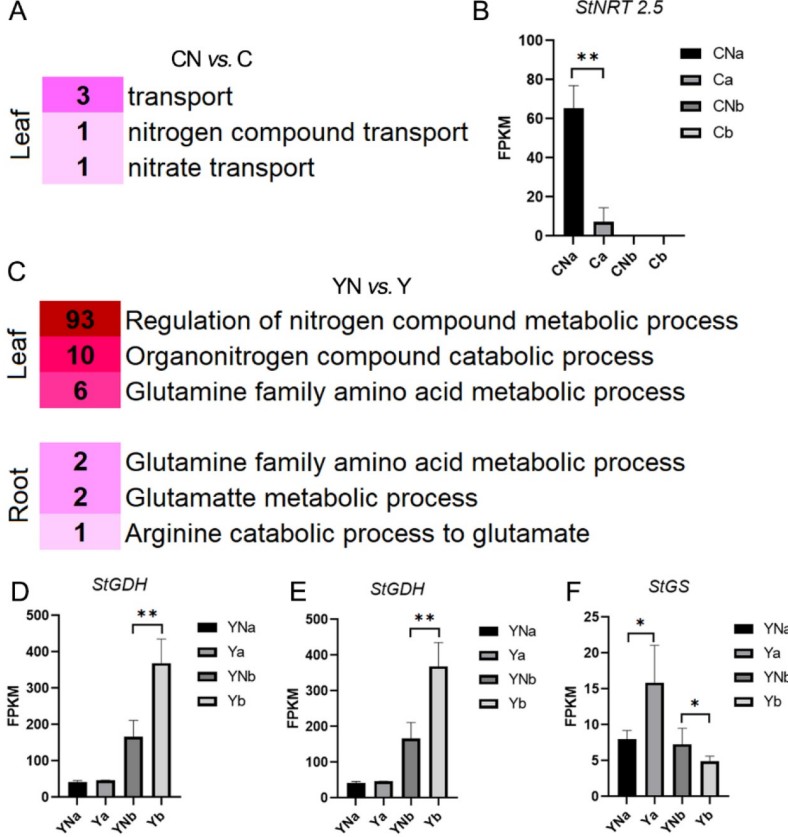

**Fig 5. The biological processes associated with N metabolism in the leaf and root in three cultivars in response to N deficiency.** A, the biological processes associated with N metabolism in the leaf of cultivar C in response to N deficiency. B, the biological processes associated with N metabolism in the leaf and root of cultivar Y in response to N deficiency. The number ahead of the processes indicated gene number enriched in this process. N = 3, ** and * means p<0.01 and p<0.05, respectively. A and b are short for leaf and root, respectively. Y, X and C indicate the cultivar of "Yanshu 4", "Xiabodi" and "Chunshu 4", respectively. N means treatment with N containing fertilizer (control). For example, YN and Y means "Yanshu 4" treated with N-containing and N-deficiency fertilizer, respectively.

DEGs between cultivars under N deficiency conditions were fewer than that of under normal fertilization conditions in both leaf and root tissues (Figs 4 and 6). These DEGs were mainly involved in the processes of response to stresses and defense. These data showed that N deficiency induced stresses in all cultivars.

### 3.6. Validation of the N-metabolism related DEGs in leaf and root in response to N deficiency

The expression of *StNRTs* (2.4, 2.5 and 2.7), *StNR*, *StNiR*, *StGDH*, *StGS* and glutamine oxoglutarate aminotransferase (*StGOGAT*) in the leaf and root of the three cultivars were validated using qRT-PCR analysis (Fig 7). We confirmed that *StNRT2.4* was increased in the leaf and root of all cultivars in response to N deficiency. However, *StNRT2.5* and *St NRT2.7* in leaf were down-regulated by N deficiency. In the root, *StNRT2.5* was up-regulated and *StNRT2.7* was down-regulated in response to N deficiency, respectively. Two transcripts of *StNiR* (PGSC0003DMG400008262 and PGSC0003DMG400025823) had similar up-regulated profiles in the leaf, and were basically down-regulated in the root in response to N deficiency. Interestingly, two *GDH* transcripts had inverse expression profiles in response to N deficiency,

**Table 1. The biological processes associated with the differentially expressed genes (DEGs) in the leaf of cultivar X versus cultivar C under N fertilization.**

| Term | ID | Input number | P-Value |
|---|---|---|---|
| **Up-regulated DEGs** | | | |
| Photosynthesis | sot00195 | 54 | 4.99E-10 |
| Porphyrin and chlorophyll metabolism | sot00860 | 28 | 1.97E-06 |
| Carbon fixation in photosynthetic organisms | sot00710 | 38 | 7.38E-06 |
| Glyoxylate and dicarboxylate metabolism | sot00630 | 34 | 7.50E-06 |
| Carbon metabolism | sot01200 | 78 | 0.000287 |
| Biosynthesis of amino acids | sot01230 | 72 | 0.000397 |
| Glycine, serine and threonine metabolism | sot00260 | 30 | 0.000675 |
| Photosynthesis-antenna proteins | sot00196 | 17 | 0.001225 |
| Sulfur metabolism | sot00920 | 17 | 0.003874 |
| Fructose and mannose metabolism | sot00051 | 23 | 0.005709 |
| Glycolysis/Gluconeogenesis | sot00010 | 39 | 0.006352 |
| Metabolic pathways | sot01100 | 439 | 0.00729 |
| Biosynthesis of secondary metabolites | sot01110 | 242 | 0.008174 |
| One carbon pool by folate | sot00670 | 10 | 0.008268 |
| Carotenoid biosynthesis | sot00906 | 15 | 0.012769 |
| Zeatin biosynthesis | sot00908 | 15 | 0.015266 |
| Cyanoamino acid metabolism | sot00460 | 16 | 0.044154 |
| **Down-regulated DEGs** | | | |
| Mismatch repair | sot03430 | 21 | 4.03E-07 |
| Homologous recombination | sot03440 | 21 | 8.13E-06 |
| DNA replication | sot03030 | 20 | 1.25E-05 |
| Nucleotide excision repair | sot03420 | 21 | 5.87E-05 |
| beta-Alanine metabolism | sot00410 | 19 | 0.000597 |
| Plant-pathogen interaction | sot04626 | 39 | 0.001005 |
| Tryptophan metabolism | sot00380 | 11 | 0.003253 |
| Plant hormone signal transduction | sot04075 | 48 | 0.005196 |
| alpha-Linolenic acid metabolism | sot00592 | 13 | 0.006505 |
| Nicotinate and nicotinamide metabolism | sot00760 | 7 | 0.007204 |
| Protein processing in endoplasmic reticulum | sot04141 | 41 | 0.009618 |
| Carotenoid biosynthesis | sot00906 | 10 | 0.011613 |
| Isoquinoline alkaloid biosynthesis | sot00950 | 8 | 0.017799 |
| Valine, leucine and isoleucine degradation | sot00280 | 14 | 0.017952 |
| Ribosome biogenesis in eukaryotes | sot03008 | 17 | 0.024065 |
| Propanoate metabolism | sot00640 | 9 | 0.027304 |
| Monoterpenoid biosynthesis | sot00902 | 4 | 0.047538 |

one of which (PGSC0003DMG400008356) was up-regulated in the leaf and root of the three cultivars by N deficiency, but the other one (PGSC0003DMG400016001) showed a reverse pattern (Fig 7). Differential expression profiles were identified in transcripts coding GS and GOGAT. These results showed that the leaf and root of potato have different responses to N deficiency (Fig 7).

## 4. Discussion

Potato is the largest non-cereal food crop worldwide and ranked as the world's fourth most important food crop after rice, wheat, and maize. In a 100-gram portion, potato could provide

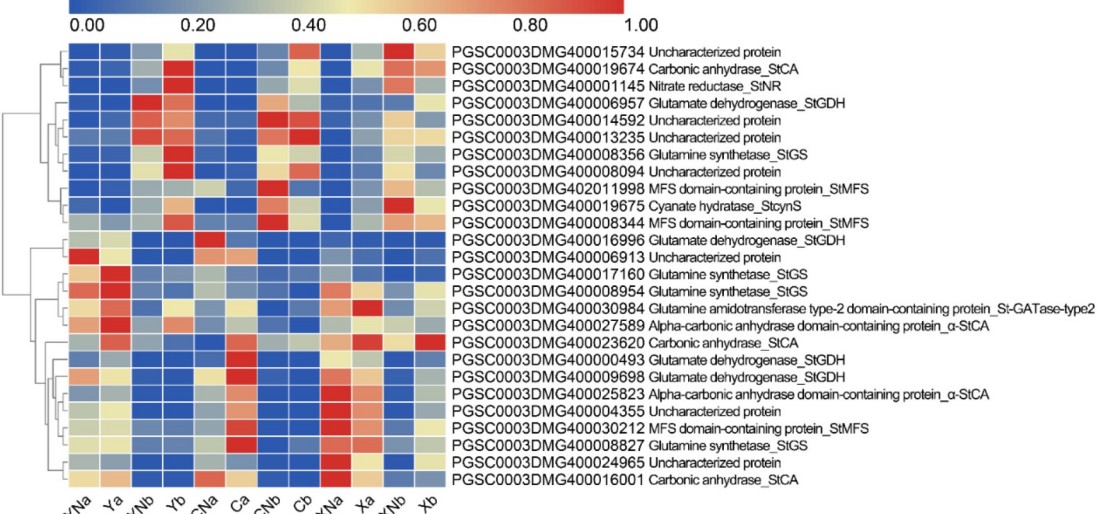

**Fig 6. The expression profiles of the genes related to N metabolism in the leaf and root in response to N deficiency.** A and B shows the expression profile of N metabolism-associated genes in the leaf and root tissue, respectively. The redder the color, the higher the amount of expression; each of the different color blocks on the left represents a class of transcript clusters with similar expression levels. A and b are short for leaf and root, respectively. Y, X and C indicate the cultivar of "Yanshu 4", "Xiabodi" and "Chunshu 4", respectively. N means treatment with N containing fertilizer (control). For example, YN and Y means "Yanshu 4" treated with N-containing and N-deficiency fertilizer, respectively.

322 kilojoules of food energy and is a rich source of vitamin B6 and vitamin C (https://ndb.nal. usda.gov/ndb/search/list). N is an essential element for potato cultivation. Different varieties have different utilization efficiency of N fertilizer. The utilization efficiency of N of "Yanshu 4", "Xiabodi" (cv. Shepody) and "Chunshu 4" potatoes are quite different in the production practice. Therefore, these three potato varieties are used as experimental objects.

Nitrogen is the nutrient that's most essential to plant growth. Plants use nitrogen to create their structure and to perform a range of functions. This chemical element is found in plant tissue, fruit, seeds and grains, and it forms part of chlorophyll, which is what makes plants green and allows them to process light into sugars [26]. N deficiency inhibited the growth and reduced the production in all the three potato cultivars. Our present study confirmed that N deficiency decreased the leaf weight, stem weight and the tuber production significantly. "Yanshu 4" had the highest production under both N-complete and N-deficient condition, followed by "Xiabodi" and "Chunshu 4". We found that plants under N deficiency had higher UPE and NUE, but lower UTE compared with N-complete fertilization, which was consistent with Dai and Zhang's reports [27,28].

Studies have shown that the root traits were important for N uptake. Commonly, the high N efficiency cultivars had higher root absorption, length, surface area, and volume than the low N efficiency cultivars, indicating that the above-mentioned root traits have improved dry matter production capacity under low nitrogen stress [27]. In the present study, N efficiencies were different among different cultivars: "Yanshu 4" had the highest UTE under N deficiency, followed by "Chunshu 4", indicating that genetic factors play an extremely important role in NUE. Transcriptome analysis showed that the leaf and root of cultivar "Yanshu 4" had the largest number of DEGs in response to N-deficiency stress compared with the other two cultivars, indicating that "Yanshu 4" is sensitive to nitrogen. An adequate supply of nitrogenous fertilizer may have excellent effects on "Yanshu 4". Additionally, Jiao et al also identified "Yanshu 4" as a N-sensitive plant [29], which further confirmed our results.

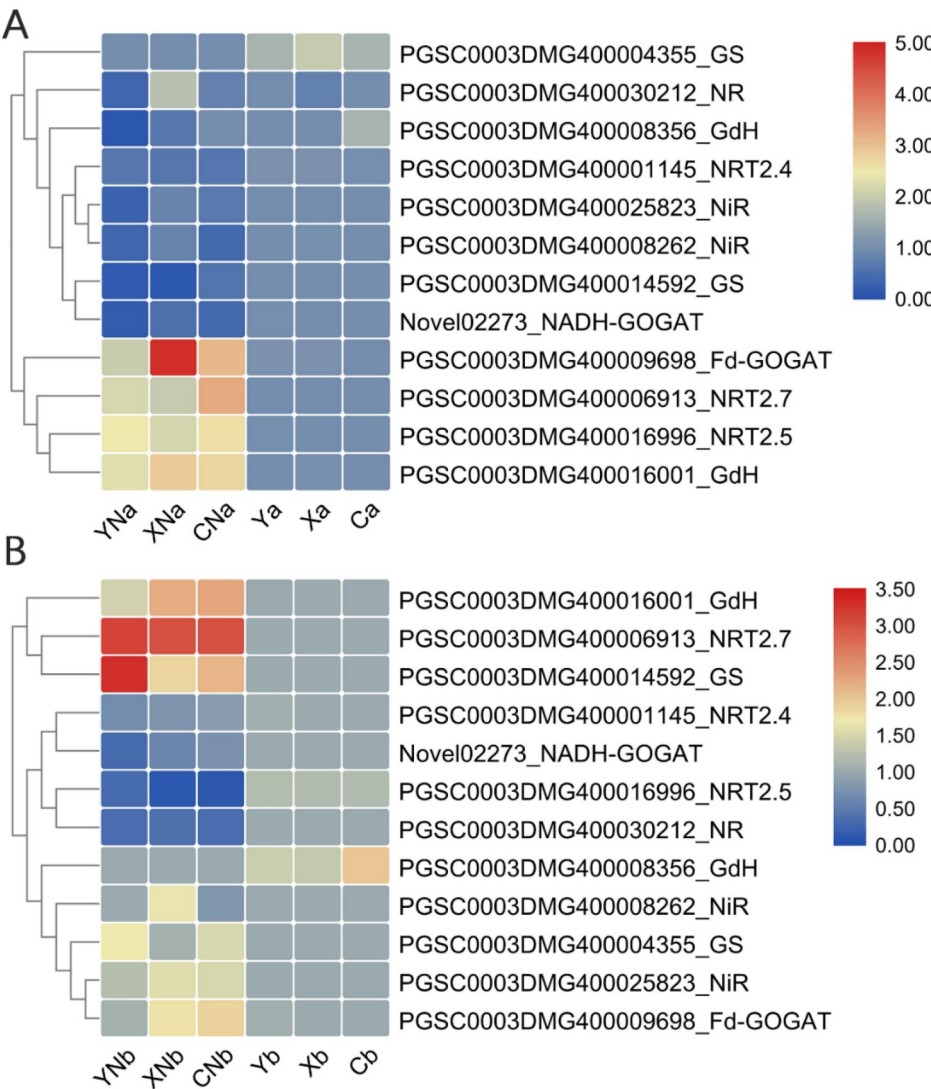

**Fig 7. The expression profiles of the genes related to N metabolism in the leaf and root in response to N deficiency.** A and B shows the expression profile of N metabolism-associated genes in the leaf and root tissue, respectively. The redder the color, the higher the amount of expression; each of the different color blocks on the left represents a class of transcript clusters with similar expression levels. A and b are short for leaf and root, respectively. Y, X and C indicate the cultivar of "Yanshu 4", "Xiabodi" and "Chunshu 4", respectively. N means treatment with N containing fertilizer (control). For example, YN and Y means "Yanshu 4" treated with N-containing and N-deficiency fertilizer, respectively.

In these three varieties of our results, the different biochemical and gene expression profiles in response to nitrogen deficiency were consistent with previous reports [30–32]. Jozefowicz et al [30] found that there are proteomic differences between two potato cultivars in response to N deficiency. Also, they reported the different activation of the GS/GOGAT pathway between the two potato cultivars. Interestingly, Jozefowicz et al found that the *GDH* was up-regulated in both the N-deficiency tolerant and sensitive potato cultivars under N-deficiency conditions, while *GS* was up-regulated only in the N tolerant potato cultivar by N deficiency [30]. Tiwari's report concluded that many potential genes play very crucial roles in N stress tolerance [32]. The difference is that this study used three potato varieties with different

resistance to N deficiency for transcriptome studies. According to the present study, Yanshu4 increased *StGAD*, *StGDH* and *StGS*, and reduced *StNRT2.5* and *StNRT2.7* in response to N deficiency. The regulation of these genes in the other two varieties was not obvious. Jozefowicz's research showed similar results with our findings [30], but the difference is that the N deficiency has a more extensive influence on the gene expression profile than protein level of high-resistant varieties. In comparison with sufficient N fertilization, N deficiency induced a large number of DEGs in the leaf and root of cultivar "Yanshu 4". However, only a few DEGs were identified in the cultivar "Xiabodi" and "Chunshu 4". Different varieties of potato showed different gene expression profiles to N stress. Two possible causes were considered, one is that there might be a lot of frontloaded genes in the two cultivars as has been found in other species [33,34]; another possibility is that they were less affected by N deficiency. Although the former seems more credible, further research is needed to verify. These results suggested that the mechanisms in potato in response to N deficiency were cultivar-dependent. At the genetic level, the different expression patterns of genes in response to N deficiency were determined by gene diversity, which might also be the root cause of different varieties of potatoes having different responses to N deficiency. How to make good use of these excellent genetic resources for cross breeding was very worthy of our future research, and "Yanshu 4" might be an excellent candidate breeding resource.

Of all these DEGs response to N deficiency in "Yanshu 4", some key genes (associated with N metabolism) were up-regulated in both leaf and root, such as *StGAD*, *StGDH*, *StGS*. Also, some down-regulated DEGs like *StNRT2.5* and *StNRT2.7* were also act as key regulators. This result was in line with the study of Gálvez et al [31], who identified 39 common N responsive genes in response to N deficiency, including *NiR*, *GOGAT*, *GS* and *GAD*. It has been reported that the increased *GS* is correlated with elevated grain yield and NUE [35,36]. *StGS* is an essential enzyme crucial for ammonium assimilation and N remobilization. It assimilates ammonium into the amide position of glutamine (Fig 8) [37]. In addition, the increased NR activity also correlated with enhanced NUE in cotton under N deficiency [38,39]. Our biochemical experiments also showed the positive relationship between the NR activity and NUE. The NR activity was decreased by N deficiency, and the decreased UTE was in line with the decreased

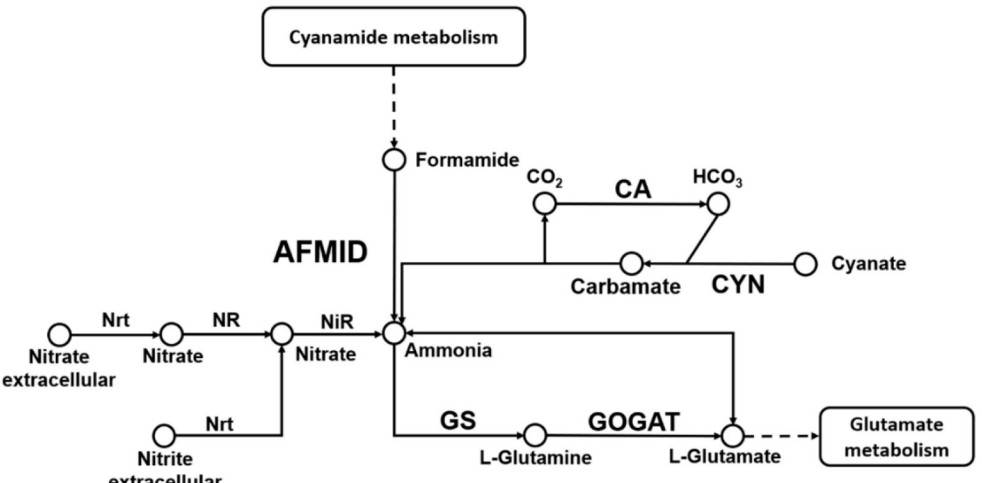

**Fig 8. A schematic representation of the N metabolism pathway emphasizing the differentially expressed genes in potato.** NRT, nitrate transporter; NR, nitrate reductase; NiR, nitrite reductase; GS, glutamine synthetase; GOGAT, glutamate synthase; AFMID, formamidase; CA, carbonic anhydrase; CYN, cyanate hydratase; GDH, glutamate synthase.

NR activity. Moreover, the increased *StGS* gene in response to N deficiency was correlated with the increased UTE in "Yanshu 4", but contrary to the decreased NUE and UPE. These data indicated that the NUE and growth of potato under N deficiency was not unilaterally determined by these two factors.

Our present study identified that several nitrate transporter coding genes were down-regulated by N deficiency including *StNRT2.5*, *StNRT2.7*, and *StNiR* in potato leaf, while *StNRT2.5* in root were up-regulated after N-deficiency treatment. These DEGs were differentially expressed among cultivars, the difference in promoter or copy number might be the two key reasons for the differential response and regulation of in three cultivars of potato. Tiwari et al reported that most of the *StNRT* family members were down-regulated in roots under low N conditions [32], which was consisted with our results. The *NRTs*, *NR* and *NiR* are crucial for the acquiring N and its conversion to ammonia (Fig 8) [40]. *NRT2* family is known to control N uptake and transport and is widely distributed in plants [41]. Lezhneva et al [40] reported that the *AtNRT2.5* was expressed in the shoot and root of *Arabidopsis* in response to N deficiency. It plays a role in obtaining N elements [40]. *Arabidopsis* has seven *NRT2* family members, and *NRT2.7* is the only *NRT2* member located on the tonoplast membrane in the seeds, and it interacts with *NAR2.1* during nitrate transport [42,43]. However, the expression profiles of *StNRT2.4*, *StNiR*, and *StNR* were increased in potato leaf by N deficiency, which suggested the increased N metabolism.

## 5. Conclusions

Nitrogen deficiency induced a large number of DEGs in the leaf and root of cultivar "Yanshu 4", and only few DEGs in the cultivar "Xiabodi" and "Chunshu 4". The mechanisms in potato in response to N deficiency were cultivar-dependent. "Yanshu 4" might be an excellent candidate cross breeding resource. The up-regulated DEGs related to N metabolism were correlated with increased UTE in potato in response to N deficiency, but the association of them with NUE needs further investigation. In addition, the decreased *StNRT2.4*, *StNiR*, and *StNR* by N deficiency indicated the increased N metabolism in potato. This study will provide data support for the breeding of high N utilization efficiency potatoes in Northeast China.

## Supporting information

**S1 Table. The sequences of primers used for the PCR analysis.**
(DOCX)

**S2 Table. RNA-seq data summary and quality analysis.**
(DOCX)

**S3 Table. Several key enzymes differentially expressed by N deficiency and between cultivars and associated with the N metabolism.**
(DOCX)

## Author Contributions

**Data curation:** Jingying Zhang, Yanfei Zhao, Yun Zhang, Yuzhu Han.

**Formal analysis:** Jingying Zhang, Yaping Wang, Yanfei Zhao, Yun Zhang, Yuzhu Han.

**Investigation:** Jiayue Zhang, Haoran Ma.

**Methodology:** Jiayue Zhang, Haoran Ma.

**Resources:** Jiayue Zhang, Haoran Ma.

**Supervision:** Yaping Wang.

**Writing – original draft:** Jingying Zhang, Yaping Wang, Yuzhu Han.

**Writing – review & editing:** Jingying Zhang, Yuzhu Han.

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
