## [Decision Letter · Decision Letter 0]

12 May 2020

PONE-D-20-07306

Nitrogen deficiency induced alterations in leaf and root of three cultivars of potato (Solanum tuberosum L.)

PLOS ONE

Dear Dr. Han,

Thank you for submitting your manuscript to PLOS ONE. After careful consideration, we feel that it has merit but does not fully meet PLOS ONE’s publication criteria as it currently stands. Therefore, we invite you to submit a revised version of the manuscript that addresses the points raised during the review process.

We would appreciate receiving your revised manuscript by 30 May 2020. To enhance the reproducibility of your results, we recommend that if applicable you deposit your laboratory protocols in protocols.io, where a protocol can be assigned its own identifier (DOI) such that it can be cited independently in the future. For instructions see: http://journals.plos.org/plosone/s/submission-guidelines#loc-laboratory-protocols

We look forward to receiving your revised manuscript.

Kind regards,

Mayank Gururani

Academic Editor

PLOS ONE

Journal Requirements:

Reviewers' comments:

Reviewer's Responses to Questions

**Comments to the Author**

1. Is the manuscript technically sound, and do the data support the conclusions?

Reviewer #1: Partly

Reviewer #2: Partly

2. Has the statistical analysis been performed appropriately and rigorously? 

Reviewer #1: Yes

Reviewer #2: Yes

3. Have the authors made all data underlying the findings in their manuscript fully available?

Reviewer #1: Yes

Reviewer #2: Yes

4. Is the manuscript presented in an intelligible fashion and written in standard English?

Reviewer #1: No

Reviewer #2: No

5. Review Comments to the Author

Reviewer #1: The study presents unique and original data that will significantly contribute to our understanding on the molecular underpinnings of nitrogen responses in plants particularly on potato. While it has already been reported by Jozefowicz et al. (2017), nitrogen deficiency does induce alterations in root proteome of potato varieties contrasting their response to low N. Just recently, Tiwari et al. (2020) presented their results on the transcriptome analysis of potato shoots, roots, and stolons under nitrogen stress. This paper by Zhang et al. investigates the association of transcriptomic profiles with the efficiencies of N using 3 potato cultivars. This study will have a bigger impact if the authors discuss the implications of their results in relation to genetic variability.

Unfortunately, the paper is poorly written. Presentation of ideas and concepts are hard to follow as it lacks the logical flow for the entire manuscript to be comprehensible. Although the experiments and statistical analysis have indeed been performed to high technical standard, however, further analysis of their data are still needed and presented for discussion and elaborated in their conclusion. Unfortunately, the conclusion is barely comprehensible as it only reiterates their findings without discussing any implications.

Also, may I suggest the title to include the term “Transcriptome” as the entire paper revolves around transcriptomic analysis of potato cultivars under nitrogen deficiency.

Reviewer #2: The manuscript by Zhang et al. investigates the role of nitrogen (N) in potato (Solanum tuberosum) plants, and in particular on three different cultivars (CVs): Yanshu 4, Xiabodi and Chunshu 4. The plants were treated with N-sufficient or with N-deficient fertilization. In the work, morphological (plant growth and tuber yield), biochemical (content soluble sugars, proteins, nitrate reductase and glutamine synthetase activities) and biomolecular parameters were evaluated.

The research is quite interesting, however some revisions are necessary before publication.

First of all, I suggest to the authors a substantial revision of the English, since there are many spelling mistakes in the text. I could not make corrections of all of them. I suggest to the author to send the manuscript to a native English speaker before submission.

At the LINE 21 – 27 of the ABSTRACT, there is a simple list of methods used in the work, without any results and discussion. Moreover, is not clear what authors mean with the sentence “and root were detected” (LINE 23). I suggest to re-write the abstract, highlighting better the results. In the present version, it is too descriptive.

The INTRODUCTION is well written, and rigidly follows the discussion of some important issues. It initially introduces the importance of potato from a nutritional and agronomic point of view, and then discusses into details the main problems linked to the cultivation of the tuber, including N levels. In this regard, the authors also describe the different practices in use for the improvement of NUE. However, I would suggest to the authors to add some additional information about the cultivars used in this work. Is there a specific reason related to the use of these selected CVs? Were they chosen for economical, nutritional, genetic or what? This information should be added to the last part of the introduction.

MATERIALS and METHODS: sometimes the methods are not well described or reported. I suppose the authors used qRT-PCR to validate the RNA-seq analysis. A couple of times they used PCR (also in the abstract) not qRT-PCR. I would suggest to add EC numbers for nitrate reductase and glutamine synthetase and also to provide some additional information about enzymatic activity measurements, although a kit was used. I don’t understand the unit used for enzyme activity calculation (see Figure 1). Moreover, what does it mean “root activity”?

In the RESULTS section, several information are missed, and must be implemented. I think it is not strictly necessary reporting numerical values in the manuscript text if present in the tables, however, at least the variation of the trend must be described (e.g. % of the change among the different CVs or between the two treatments). On the other hand, when data are expressed as Figures, it is necessary to report in the text the most important and representative numerical values (means ± standard deviation), and describe the changes among CVs and treatments.

The DISCUSSION is well motivated, but is limited only to the data obtained in this work. This part must be expanded. I suggest to introduce a short initial part, in which the importance of nutritional and economical aspects of potato are discussed in relation with the three different CVs employed. In addition, the importance of N, and its deficiency in plant nutrition has to be introduced.

The CONCLUSIONS are too synthetic and don’t highlight the obtained results. I suggest to expand also this section by explaining the importance of the research and the future applications.

LEGENDS FOR FIGURES: they must be improved. The authors should report as much information as possible. For example the statistical treatment used and the meaning of the letters on the bars (Figure 1). The legend must be clear for the reader

6. PLOS authors have the option to publish the peer review history of their article (what does this mean?). If published, this will include your full peer review and any attached files.

Reviewer #1: No

Reviewer #2: No

---

## [Author Response · Author response to Decision Letter 0]

20 Jul 2020

Dear Editor,

Thank you for your letter and for the reviewers’ comments concerning our manuscript entitled “Nitrogen deficiency induced alterations in leaf and root of three cultivars of potato (Solanum tuberosum L.)”. Those comments are all valuable. We have studied comments carefully and made corrections.

The revised manuscript is highlighted in Tracked Changes version.

Point-by-point response to the reviewer’s comments

Reviewer reports:

Reviewer 1: 

The study presents unique and original data that will significantly contribute to our understanding on the molecular underpinnings of nitrogen responses in plants particularly on potato. While it has already been reported by Jozefowicz et al. (2017), nitrogen deficiency does induce alterations in root proteome of potato varieties contrasting their response to low N. Just recently, Tiwari et al. (2020) presented their results on the transcriptome analysis of potato shoots, roots, and stolons under nitrogen stress. This paper by Zhang et al. investigates the association of transcriptomic profiles with the efficiencies of N using 3 potato cultivars. This study will have a bigger impact if the authors discuss the implications of their results in relation to genetic variability.

Unfortunately, the paper is poorly written. Presentation of ideas and concepts are hard to follow as it lacks the logical flow for the entire manuscript to be comprehensible. Although the experiments and statistical analysis have indeed been performed to high technical standard, however, further analysis of their data are still needed and presented for discussion and elaborated in their conclusion. Unfortunately, the conclusion is barely comprehensible as it only reiterates their findings without discussing any implications.

Response: Thank you for your comments. Firstly, we accept the suggestion to try our best to improve the English language. To further improve the quality of written English, the manuscript has also been edited by Caughman Corey, who was a native English speaker. Then, we have reorganized the manuscript and hope that the new manuscript will be more logical. In addition, to better present the results, we have made major revisions to the results section. New data were added. Also, we re-drawn figure 3 and added N-related gene expression results (Figure 4). Importantly, we have revised the discussion and conclusion sections, hoping that the newly submitted manuscript will be improved.

Also, may I suggest the title to include the term “Transcriptome” as the entire paper revolves around transcriptomic analysis of potato cultivars under nitrogen deficiency.

Response: Thank you for your comments. We re-titled the manuscript to “Transcriptome analysis reveals Nitrogen deficiency induced alterations in leaf and root of three cultivars of potato (Solanum tuberosum L.)”

Reviewer 2: 

First of all, I suggest to the authors a substantial revision of the English, since there are many spelling mistakes in the text. I could not make corrections of all of them. I suggest to the author to send the manuscript to a native English speaker before submission.

Response: Thank you for your comments. We accept the suggestion to try our best to improve the English language. To further improve the quality of written English, the manuscript has also been edited by Caughman Corey, who was a native English speaker.

At the LINE 21 – 27 of the ABSTRACTS, there is a simple list of methods used in the work, without any results and discussion. Moreover, is not clear what authors mean with the sentence “and root were detected” (LINE 23). I suggest to re-write the abstract, highlighting better the results. In the present version, it is too descriptive.

Response: Thank you for your comments. We re-write the abstract section and highlight the main results. The reorganized abstract is improved.

The INTRODUCTION is well written, and rigidly follows the discussion of some important issues. It initially introduces the importance of potato from a nutritional and agronomic point of view, and then discusses into details the main problems linked to the cultivation of the tuber, including N levels. In this regard, the authors also describe the different practices in use for the improvement of NUE. However, I would suggest to the authors to add some additional information about the cultivars used in this work. Is there a specific reason related to the use of these selected CVs? Were they chosen for economical, nutritional, genetic or what? This information should be added to the last part of the introduction.

Response: Thank you for your comments. The utilization efficiency of N of “Yanshu 4”, “Xiabodi” (cv. Shepody) and “Chunshu 4” potatoes are quite different in the production practice. Therefore, these three potato varieties are used as experimental objects.

MATERIALS and METHODS: sometimes the methods are not well described or reported. I suppose the authors used qRT-PCR to validate the RNA-seq analysis. A couple of times they used PCR (also in the abstract) not qRT-PCR. I would suggest to add EC numbers for nitrate reductase and glutamine synthetase and also to provide some additional information about enzymatic activity measurements, although a kit was used. I don’t understand the unit used for enzyme activity calculation (see Figure 1). Moreover, what does it mean “root activity”?

Response: Thank you for your comments. All the PCR mentioned in MATERIALS and METHODS should be qRT-PCR. We have revised the related parts. The EC number for nitrate reductase and glutamine synthetase was provide as required. The function of root activity was added in the method section. For the unit, we revised it in the newly submitted manuscript.

In the RESULTS section, several information are missed, and must be implemented. I think it is not strictly necessary reporting numerical values in the manuscript text if present in the tables, however, at least the variation of the trend must be described (e.g. % of the change among the different CVs or between the two treatments). On the other hand, when data are expressed as Figures, it is necessary to report in the text the most important and representative numerical values (means ± standard deviation), and describe the changes among CVs and treatments.

Response: Thank you for your comments. We added some main information to the manuscript. But it is unrealistic to fully describe the range of increase or decrease, especially the content in the table. We have two treatments for 3 varieties, so the range of variation is diverse. Even the changing trends of different varieties are also different between different treatments. Therefore, we only added the main content. 

The DISCUSSION is well motivated, but is limited only to the data obtained in this work. This part must be expanded. I suggest to introduce a short initial part, in which the importance of nutritional and economical aspects of potato are discussed in relation with the three different CVs employed. In addition, the importance of N, and its deficiency in plant nutrition has to be introduced.

Response: Thank you for your comments. We added the importance of nutritional and economical aspects of potato. The utilization efficiency of N of “Yanshu 4”, “Xiabodi” (cv. Shepody) and “Chunshu 4” potatoes are quite different in the production practice. Therefore, these three potato varieties are used as experimental objects. The importance of N was also discussed in the newly submitted manuscript. 

The CONCLUSIONS are too synthetic and don’t highlight the obtained results. I suggest to expand also this section by explaining the importance of the research and the future applications.

Response: Thank you for your comments. We rewrote our conclusions, summarizing the important results of this study and the importance of this study for potato breeding.

LEGENDS FOR FIGURES: they must be improved. The authors should report as much information as possible. For example the statistical treatment used and the meaning of the letters on the bars (Figure 1). The legend must be clear for the reade

Response: Thank you for your comments. All the legends of figures and tables were improved.

---

## [Decision Letter · Decision Letter 1]

17 Aug 2020

PONE-D-20-07306R1

Transcriptome analysis reveals Nitrogen deficiency induced alterations in leaf and root of three cultivars of potato (Solanum tuberosum L.)

PLOS ONE

Dear Dr. Han,

Thank you for submitting your manuscript to PLOS ONE. After careful consideration, we feel that it has merit but does not fully meet PLOS ONE’s publication criteria as it currently stands. Therefore, we invite you to submit a revised version of the manuscript that addresses the points raised during the review process.

We look forward to receiving your revised manuscript.

Kind regards,

Mayank Gururani

Academic Editor

PLOS ONE

Reviewers' comments:

Reviewer's Responses to Questions

**Comments to the Author**

1. If the authors have adequately addressed your comments raised in a previous round of review and you feel that this manuscript is now acceptable for publication, you may indicate that here to bypass the “Comments to the Author” section, enter your conflict of interest statement in the “Confidential to Editor” section, and submit your "Accept" recommendation.

Reviewer #1: (No Response)

Reviewer #2: All comments have been addressed

2. Is the manuscript technically sound, and do the data support the conclusions?

Reviewer #1: Yes

Reviewer #2: Yes

3. Has the statistical analysis been performed appropriately and rigorously? 

Reviewer #1: No

Reviewer #2: Yes

4. Have the authors made all data underlying the findings in their manuscript fully available?

Reviewer #1: Yes

Reviewer #2: Yes

5. Is the manuscript presented in an intelligible fashion and written in standard English?

Reviewer #1: No

Reviewer #2: Yes

6. Review Comments to the Author

Reviewer #1: This paper by Zhang et al. investigates the association of transcriptomic profiles with N deficiency using 3 potato cultivars. This study will have a bigger impact if the authors discuss the implications of their results in relation to genetic variability which was barely discussed. Similarly, it is best to include in their discussion on how their results differ or add up to those already contributed by Jozefowicz et al. (2017) and Tiwari et al. (2020). Moreover, the conclusion needs to be further improved and expounded as it remains shallow reiterating their findings but without value/importance.

Statistical analyses only mentioned unpaired t-test and two-way ANOVA in the Methods while there is a plethora of statistical analyses (Holm- Sidak, Tukey’s Test, etc) which are more appropriate for this study. How was the correlation analysis done to conclude positive/negative correlation?

The paper needs to be presented in a more intelligible manner (scientific jargon not just standard English). The following lines/sentences needs to be rephrased or improved:

Lines: 49, 69, 74-75, 94-97, 102-104, 179-188, 199-201, 208-214, 220-225,228-231, 237-239, 243-251,270-271, 275-276, 284-288, 309-311, 354, 370, 374-376

Others:

Line 28 what is correlation relationship? Isn’t this positive or negative correlation?

Line 79 Delete “in April 2017”

Line 110 RNA-seq analysis using ___ (brand & model of equipment)

Line 111 delete as YNa, YNb …..

Line 216 Delete the terms “statistics of the”

Tables 1 and 2 need to be presented as graphs in figure with error bars representing SD.

Line 251 N containing fertilizer (not contained)

Line 299-300 what is PGSC… and PGSC…?

Line 323 start new paragraph with, “Nitrogen is the nutrient…

Line 330 what is N-colplete fertilization?

Line 331 start new paragraph with, “Studies have shown ….

Reviewer #2: The manuscript has been improved and most part of my comments were properly addressed.

However, some revision is still necessary.

Line 28: GS and NR must be written with the full names as well. The same for StGDH, StGS and StCA (Line 30) and MFS members, like StNRT2.4, StNRT2.5 and StNRT2.7 (Line 33)

Line 50. Its tuber is used not tuber of which

Line 116. I would write the anthrone method and Coomassie Brilliant Blue G-250 dye-binding method were used…

Line 117. To detect not to detected

Line 118. I think that it is better to report the plural. Soluble sugars and soluble proteins.

Line 125: which kind of phosphate buffer? K or Na? I think the following sentence is unclear: Then the color reaction is carried out, and finally the absorbance is measured to determine the enzyme activity. Better to rewrite the sentence and use the past tense.

Line 132. RNAseq analysis since seq is already sequencing

Line 164. CTAB: report the full name as well. Methods should be singular, so method

Line 207. Sugars and proteins, not sugar and protein

Line 243. Respectively, not respectiverly

Line 261. Figure legend. Bar not bat

Line 270. DEG not DEGs

Line 318. N metabolism not N metabolisms

Line 392. N-completed not N-colplete

Line 416. DEG not DEGs

Supplementary

Table S1: qRT-PCR not PCR

7. PLOS authors have the option to publish the peer review history of their article (what does this mean?). If published, this will include your full peer review and any attached files.

Reviewer #1: No

Reviewer #2: No

---

## [Author Response · Author response to Decision Letter 1]

30 Aug 2020

Dear editor:

Thank you for your letter and for the reviewers’ comments concerning our manuscript entitled “Transcriptome analysis reveals Nitrogen deficiency induced alterations in leaf and root of three cultivars of potato (Solanum tuberosum L.)”. Those comments are all valuable. We have studied comments carefully and made corrections.

The revised manuscript is highlighted in Tracked Changes version.

Point-by-point response to the reviewer’s comments

Response:

This paper by Zhang et al. investigates the association of transcriptomic profiles with N deficiency using 3 potato cultivars. This study will have a bigger impact if the authors discuss the implications of their results in relation to genetic variability which was barely discussed. 

Response: Thank you for your comments. The discussion section was improved. At the genetic level, the different expression patterns of genes in response to N deficiency were determined by gene diversity, which might also be the root cause of different varieties of potatoes having different responses to N deficiency. How to make good use of these excellent genetic resources for cross breeding was very worthy of our future research, and “Yanshu 4” might be an excellent candidate breeding resource. This part of the content has been added in the discussion.

Similarly, it is best to include in their discussion on how their results differ or add up to those already contributed by Jozefowicz et al. (2017) and Tiwari et al. (2020). 

Response: Thank you for your comments. Jozefowicz’s research showed similar results with our findings, but the difference is that the N deficiency has a more extensive influence on the gene expression profile than protein level of high-resistant varieties. The experimental design of Tiwari’s research is also different from this study, but some of the results are consistent. In the discussion, we added a part of the content to explain the difference between this study and previous studies, but more importantly, discuss the possible reasons that caused the results of this study

Moreover, the conclusion needs to be further improved and expounded as it remains shallow reiterating their findings but without value/importance.

Response: Thank you for your comments. We have condensed the conclusion part to make the conclusion more prominent.

Statistical analyses only mentioned unpaired t-test and two-way ANOVA in the Methods while there is a plethora of statistical analyses (Holm- Sidak, Tukey’s Test, etc) which are more appropriate for this study. 

Response: Thank you for your comments. The data were analyzed using the ANOVA test followed by Tukey post-hoc. We missed the test method in the previous version, this time we added it.

How was the correlation analysis done to conclude positive/negative correlation?

Response: Thank you for your comments. Correlation were analysis by Pearson correlation analysis.

The paper needs to be presented in a more intelligible manner (scientific jargon not just standard English). The following lines/sentences needs to be rephrased or improved:

Lines: 49, 69, 74-75, 94-97, 102-104, 179-188, 199-201, 208-214, 220-225,228-231, 237-239, 243-251,270-271, 275-276, 284-288, 309-311, 354, 370, 374-376

Response: Thank you for your comments. All the mentioned parts were revised as required.

Others:

Line 28 what is correlation relationship? Isn’t this positive or negative correlation?

Line 79 Delete “in April 2017”

Line 110 RNA-seq analysis using ___ (brand & model of equipment)

Line 111 delete as YNa, YNb …..

Line 216 Delete the terms “statistics of the”

Tables 1 and 2 need to be presented as graphs in figure with error bars representing SD.

Line 251 N containing fertilizer (not contained)

Line 299-300 what is PGSC… and PGSC…?

Line 323 start new paragraph with, “Nitrogen is the nutrient…

Line 330 what is N-colplete fertilization?

Line 331 start new paragraph with, “Studies have shown ….

Response: Thank you for your comments. The mentioned parts were all revised as required.

Reviewer #2: The manuscript has been improved and most part of my comments were properly addressed.

However, some revision is still necessary.

Line 28: GS and NR must be written with the full names as well. The same for StGDH, StGS and StCA (Line 30) and MFS members, like StNRT2.4, StNRT2.5 and StNRT2.7 (Line 33)

Line 50. Its tuber is used not tuber of which

Line 116. I would write the anthrone method and Coomassie Brilliant Blue G-250 dye-binding method were used…

Line 117. To detect not to detected

Line 118. I think that it is better to report the plural. Soluble sugars and soluble proteins.

Line 125: which kind of phosphate buffer? K or Na? I think the following sentence is unclear: Then the color reaction is carried out, and finally the absorbance is measured to determine the enzyme activity. Better to rewrite the sentence and use the past tense.

Line 132. RNAseq analysis since seq is already sequencing

Line 164. CTAB: report the full name as well. Methods should be singular, so method

Line 207. Sugars and proteins, not sugar and protein

Line 243. Respectively, not respectiverly

Line 261. Figure legend. Bar not bat

Line 270. DEG not DEGs

Line 318. N metabolism not N metabolisms

Line 392. N-completed not N-colplete

Line 416. DEG not DEGs

Response: Thank you for your approval and comments. All the mentioned parts in the manuscript was revised as required.

Supplementary

Table S1: qRT-PCR not PCR

Response: Thank you for your comments. We revised it as required.

---

## [Decision Letter · Decision Letter 2]

14 Sep 2020

PONE-D-20-07306R2

Transcriptome analysis reveals Nitrogen deficiency induced alterations in leaf and root of three cultivars of potato (Solanum tuberosum L.)

PLOS ONE

Dear Dr. Han,

Thank you for submitting your manuscript to PLOS ONE. After careful consideration, we feel that it has merit but does not fully meet PLOS ONE’s publication criteria as it currently stands. Therefore, we invite you to submit a revised version of the manuscript that addresses the points raised during the review process.

We look forward to receiving your revised manuscript.

Kind regards,

Mayank Gururani

Academic Editor

PLOS ONE

Reviewers' comments:

Reviewer's Responses to Questions

**Comments to the Author**

1. If the authors have adequately addressed your comments raised in a previous round of review and you feel that this manuscript is now acceptable for publication, you may indicate that here to bypass the “Comments to the Author” section, enter your conflict of interest statement in the “Confidential to Editor” section, and submit your "Accept" recommendation.

Reviewer #2: All comments have been addressed

Reviewer #3: (No Response)

2. Is the manuscript technically sound, and do the data support the conclusions?

Reviewer #2: Yes

Reviewer #3: (No Response)

3. Has the statistical analysis been performed appropriately and rigorously? 

Reviewer #2: Yes

Reviewer #3: (No Response)

4. Have the authors made all data underlying the findings in their manuscript fully available?

Reviewer #2: Yes

Reviewer #3: (No Response)

5. Is the manuscript presented in an intelligible fashion and written in standard English?

Reviewer #2: Yes

Reviewer #3: (No Response)

6. Review Comments to the Author

Reviewer #2: (No Response)

Reviewer #3: Overall the paper is well-written, and provides valuable insight on transcriptome analysis reveals N-deficiency in three cultivars of potato (Solanum tuberosum L.). Below are suggestions and comments to improve the clarity and message of the manuscript.

>1.The authors choose three cultivars potato, Yanshu 4, Xiabodi and Chunshu 4. How did the authors decide three cultivars? From cultivar resources?

There was different genetic background in three cultivars potato, the difference in the growth performance parameters may be not only affected by N-related genes. So, the hybrids offspring of two different varieties were more reasonable.

>2. Unclear statistics: you mention three plants for each replicate have been used, the number of n should be used for statistical evaluation, which tests were applied in legends?

>3.In fig4 C D, why the DEGs were less under N-deficiency condition in cultivars. There may be a lot frontloaded genes in cultivars.

Reference: Barshis etal, 2013 “Genomic basis for coral resilience to climate change”

>4. In RNA-seq analysis: why choose the second leaf? not flag leaf? And in RT-qPCR, which leaf was used, growth stage?

>5. In Fig6 and 7, It would be two reasons about the differential response and regulation of in three cultivars of potato, promoter or copy number. The expression of one gene was not usually compared among different cultivars by RT-qPCR.

7. PLOS authors have the option to publish the peer review history of their article (what does this mean?). If published, this will include your full peer review and any attached files.

Reviewer #2: No

Reviewer #3: No

---

## [Author Response · Author response to Decision Letter 2]

29 Sep 2020

Dear editor:

Thank you for your letter and for the reviewers’ comments concerning our manuscript entitled “Transcriptome analysis reveals Nitrogen deficiency induced alterations in leaf and root of three cultivars of potato (Solanum tuberosum L.)”. Those comments are all valuable. We have studied comments carefully and made corrections.

The revised manuscript is highlighted in Tracked Changes version.

Point-by-point response to the reviewer’s comments

Review Comments to the Author

Reviewer #2: (No Response)

Reviewer #3: Overall the paper is well-written, and provides valuable insight on transcriptome analysis reveals N-deficiency in three cultivars of potato (Solanum tuberosum L.). Below are suggestions and comments to improve the clarity and message of the manuscript.

>1.The authors choose three cultivars potato, Yanshu 4, Xiabodi and Chunshu 4. How did the authors decide three cultivars? From cultivar resources? 

There was different genetic background in three cultivars potato, the difference in the growth performance parameters may be not only affected by N-related genes. So, the hybrids offspring of two different varieties were more reasonable.

Response: Thank you for your comments. In the production practice, we found that the utilization efficiency of N of “Yanshu 4”, “Xiabodi” (cv. Shepody) and “Chunshu 4” potatoes are quite different in the production practice. According to production experience, under the condition of sufficient nitrogen fertilizer, “Yanshu 4” was a high-absorption but low-utilization potato, “Xiabodi” was a medium-absorption and medium-utilization potato, and “Chunshu 4” was a low-absorption but high-utilization potato (lines 72-75). The aim of this study was to reveal the transcriptional responses of different cultivars to N deficiency under different conditions.

In addition, you encourage us to hybrid offspring of two different varieties, this is very important for our follow-up study. We are willing to accept your opinion to carry out hybridization experiment and reveal the location of functional genes and QTLs associated with the N-utilization efficiency. 

Thanks again for the above comment and suggestion.

>2. Unclear statistics: you mention three plants for each replicate have been used, the number of n should be used for statistical evaluation, which tests were applied in legends?

Response: Thank you for your comments. We added the number of replicates in the statistical analysis section. Also, the tests were also added in the legends of Figure 1, 2 and 3. 

>3. In fig4 C D, why the DEGs were less under N-deficiency condition in cultivars. There may be a lot frontloaded genes in cultivars.

Reference: Barshis etal, 2013 “Genomic basis for coral resilience to climate change”

Response: Thank you for your comments. We accept your suggestion and discuss it in the revised manuscript (Discussion section, lines 361-362).

>4. In RNA-seq analysis: why choose the second leaf? not flag leaf? And in RT-qPCR, which leaf was used, growth stage?

Response: Thank you for your comments. Because of individual differences, the flag leaf size is different. To avoid the impact of this difference, we used the second leaf. For qRT-PCR analysis, the second leaf was also used. Samples were collected at bud stage (line 149).

>5. In Fig6 and 7, It would be two reasons about the differential response and regulation of in three cultivars of potato, promoter or copy number. The expression of one gene was not usually compared among different cultivars by RT-qPCR.

Response: Thank you for your comments. We added the reason in discussion section. Also, we have modified the figures to reflect intra species comparisons rather than inter variety comparisons (lines 379-379). Thank you again for your comments, which is very helpful to improve our article.

---

## [Decision Letter · Decision Letter 3]

1 Oct 2020

Transcriptome analysis reveals Nitrogen deficiency induced alterations in leaf and root of three cultivars of potato (Solanum tuberosum L.)

PONE-D-20-07306R3

Dear Dr. Han,

We’re pleased to inform you that your manuscript has been judged scientifically suitable for publication and will be formally accepted for publication once it meets all outstanding technical requirements.

Kind regards,

Mayank Gururani

Academic Editor

PLOS ONE

Additional Editor Comments (optional):

Reviewers' comments:

Reviewer's Responses to Questions

**Comments to the Author**

1. If the authors have adequately addressed your comments raised in a previous round of review and you feel that this manuscript is now acceptable for publication, you may indicate that here to bypass the “Comments to the Author” section, enter your conflict of interest statement in the “Confidential to Editor” section, and submit your "Accept" recommendation.

Reviewer #3: (No Response)

2. Is the manuscript technically sound, and do the data support the conclusions?

Reviewer #3: Yes

3. Has the statistical analysis been performed appropriately and rigorously? 

Reviewer #3: Yes

4. Have the authors made all data underlying the findings in their manuscript fully available?

Reviewer #3: Yes

5. Is the manuscript presented in an intelligible fashion and written in standard English?

Reviewer #3: Yes

6. Review Comments to the Author

Reviewer #3: (No Response)

7. PLOS authors have the option to publish the peer review history of their article (what does this mean?). If published, this will include your full peer review and any attached files.

Reviewer #3: No

---

## [Editor Report · Acceptance letter]

15 Oct 2020

PONE-D-20-07306R3 

Transcriptome analysis reveals Nitrogen deficiency induced alterations in leaf and root of three cultivars of potato (*Solanum* *tuberosum* L.) 

Dear Dr. Han:

I'm pleased to inform you that your manuscript has been deemed suitable for publication in PLOS ONE. Congratulations! Your manuscript is now with our production department. 

Kind regards, 

on behalf of

Dr. Mayank Gururani 

Academic Editor

PLOS ONE